# Efficient Uptake of Angiotensin-Converting Enzyme II Inhibitor Employing Graphene Oxide-Based Magnetic Nanoadsorbents

**Miguel Pereira de Oliveira [1], Carlos Schnorr [2], Theodoro da Rosa Salles [1], Franciele da Silva Bruckmann [3], Luiza Baumann [3], Edson Irineu Muller [3], Wagner Jesus da Silva Garcia [4], Artur Harres de Oliveira [5], Luis F. O. Silva [2] and Cristiano Rodrigo Bohn Rhoden [2,6,*]**

[1] Laboratório de Materiais Magnéticos Nanoestruturados—LaMMaN, Universidade Franciscana, Santa Maria 97010-030, Brazil

[2] Department of Civil and Environmental, Universidad de la Costa, CUC, Calle 58 # 55–66, Barranquilla 080002, Colombia

[3] Programa de Pós-graduação em Química, Universidade Federal de Santa Maria—UFSM, Santa Maria 97105-900, Brazil

[4] Departamento de Desenho Industrial, Universidade Federal de Santa Maria—UFSM, Santa Maria 97105-900, Brazil

[5] Departamento de Física, Universidade Federal de Santa Maria—UFSM, Santa Maria 97105-900, Brazil

[6] Programa de Pós-graduação em Nanociências, Universidade Franciscana—UFN, Santa Maria 97010-030, Brazil

* Correspondence: cristianorbr@gmail.com

**Abstract:** This paper reports a high efficiency uptake of captopril (CPT), employing magnetic graphene oxide (MGO) as the adsorbent. The graphene oxide (GO) was produced through an oxidation and exfoliation method, and the magnetization technique by the co-precipitation method. The nanomaterials were characterized by FTIR, XRD, SEM, Raman, and VSM analysis. The optimal condition was reached by employing $GO \cdot Fe_3O_4$ at pH 3.0 (50 mg of adsorbent and 50 mg $L^{-1}$ of CPT), presenting values of removal percentage and maximum adsorption capacity of 99.43% and 100.41 mg $g^{-1}$, respectively. The CPT adsorption was dependent on adsorbent dosage, initial concentration of adsorbate, pH, and ionic strength. Sips and Elovich models showed the best adjustment for experimental data, suggesting that adsorption occurs in a heterogeneous surface. Thermodynamic parameters reveal a favorable, exothermic, involving a chemisorption process. The magnetic carbon nanomaterial exhibited a high efficiency after five adsorption/desorption cycles. Finally, the $GO \cdot Fe_3O_4$ showed an excellent performance in CPT removal, allowing future application in waste management.

**Keywords:** adsorption; carbon nanomaterials; magnetite; captopril





## 1. Introduction

Water contamination by emerging pollutants (EPs) has become an important problem around the world [1]. Amongst these, drugs and personal care products are widely found in water bodies, due to their intrinsic characteristics and incomplete removal by current methods [2]. Additionally, due to their unique properties and xenobiotic-like character, these compounds present a high tendency to bioaccumulation, which leads to the development of diverse damages to human health, animals, and the environment [3]. In addition, high consumption, inadequate disposal, and incomplete metabolism are associated with the high and constant generation of waste. Due to low concentration and traces in water, the conventional systems are not able to remove these compounds, resulting in a secondary source of exposition and contamination [4].

Captopril (CPT) (1-(3-mercapto-2-D-methyl-1-oxopropyl)-L proline is a hydrophilic drug used to inhibit angiotensin-converting enzyme II (ACE II) and is one of the most commonly prescribed medicines for the treatment of high blood pressure and heart failure [5,6].

Although this medicine is effective for control of diseases, around 50% of CPT is excreted by kidneys in unchanged form, contributing to water contamination [7]. Additionally, this pharmaceutical compound may cause diverse side effects and toxicity to organisms [8,9].

In this scenario, diverse methods have been explored for EPs removal from aqueous medium, such as Fenton [10], photocatalyst [11], ozonation [12], and adsorption [13,14]. The adsorption technique is characterized by the capture of contaminant (adsorbate) using a solid surface (adsorbent). This method presents advantages compared with other processes, such as low energy requirements and easy operation, without byproducts and generation of sludge [15]. Regarding the uptake of captopril, recent studies reported its removal using alternative materials from the biomass and lignocellulosic waste [9,16]. Considering the high consumption of captopril and the potential damage to health, and the limited studies that report its removal from the aquatic environment, it is necessary to use and design alternative and novel adsorbents. Additionally, nanotechnology is an excellent tool to enhance the process, using adsorbents at nanoscale, improving the properties and characteristics of conventional adsorbents, as well as the adsorption efficiency [17,18].

Among these, carbon nanomaterials, such as graphene oxide (GO) can be used in the adsorption process, due to the high specific surface area, reactivity, and presence of oxygenated groups [19]. Nonetheless, the unique properties of GO allow combination with other nanoparticles, as magnetic nanoparticles (MNPs), conferring new properties as well as avoiding the filtration and centrifugation steps. Recently, Li et al. [20] synthesized a magnetic nanocomposite with GO and hydroxyapatite for Pb (II) removal from water. The results demonstrate that nanomaterials exhibited a high adsorption capacity and maintained efficiency after several cycles of adsorption/desorption.

This work reports captopril adsorption using magnetic graphene oxide ($GO \cdot Fe_3O_4$), synthesized through a straightforward method, using only $Fe^{2+}$ as the iron source [21]. The adsorption behavior was evaluated using different experimental variables. The isotherm and kinetic models were used to understand the adsorption equilibrium and kinetic profile. Additionally, the adsorbent regeneration was investigated to determine the lifetime and efficiency after diverse adsorption/desorption processes. The mechanistic hypotheses for captopril adsorption onto graphene oxide-based magnetic adsorbent are also proposed, based on chemical species and magnetite incorporation on the GO surface.

## 2. Materials and Methods

### 2.1. Synthesis of Graphene Oxide and Graphene Oxide-Based Magnetic Nanoadsorbent

Graphene oxide was synthesized following Salles et al. [22]. 1 g of graphite in flakes (Sigma-Aldrich®, St. Louis, MO, USA) was added to a flask with 60 mL of $H_2SO_4$ 98% (Synth®) under magnetic stirring for 10 min at room temperature. Sequentially, 6 g of $KMnO_4$ 99% (Synth®) was slowly added over 20 min. The solution was heated at 40 °C and stirred continuously for 5 h. Afterward, 180 mL of distilled $H_2O$ was slowly added and stirring for 12 h at room temperature. Sequentially, the reaction was heated at 40 °C for 2 h. Finally, 300 mL of distilled $H_2O$ and 10 mL of $H_2O_2$ 29% (Synth®) was added into the solution. The yellow solution was washed until pH 5.0 and dried in an oven (DeLeo) at 50 °C for 24 h.

The magnetic graphene oxide was produced by the co-precipitation method [21]. In a 250 mL round-bottom flask containing 120 mL of ultrapure water, 100 mg of GO were added to 1000 mg of $FeCl_2 \geq 99\%$ (Sigma-Aldrich®). Afterwards, ammonium hydroxide (Synth®) was added to reach pH 9.0. Sequentially, the mixture was submitted to ultrasonic radiation (Elma, power 150 W) for 90 min at room temperature. Finally, the product was purified with distilled water and acetone (Synth®), and dried in an oven (DeLeo) at 50 °C for total solvents evaporation.

### 2.2. Adsorbent Characterization

Graphene oxide and magnetic graphene oxide were characterized by several different techniques. The functional groups were determined through the Fourier Transform spec-

troscopy (Perkin-Elmer, model Spectro One). The crystallinity was determined by X-ray diffraction (Bruker diffractometer, model D2 Phaser), and Raman Spectroscopy (Renishaw inVia spectrometer system). The morphological characteristics were evaluated using a scanning electron microscopic (Zeiss Sigma 300 VP). The magnetic properties were evaluated with a vibrating sample magnetometer (Stanford Research Systems Model SR830 DSP lock-in amplifier, coupled to a Stanford Research Systems low-noise pre-amplifier model SR560, including a Kepco bipolar operation power supply).

The zero point of charge ($pH_{ZPC}$) of MGO was evaluated by the following procedure. 10 mg of adsorbent was added to a 100 mL flask containing 50 mL of ultrapure water (Milli-Q®, Darmstadt, Germany) and pH adjusted in the range of (2.0–12.0) using NaOH (0.01 mol $L^{-1}$) and HCl (0.01 mol $L^{-1}$). Sequentially, the samples were maintained under stirring (120 rpm) for 24 h at room temperature. Subsequentially, the final pH was measured with a potentiometer (Digimed).

### 2.3. Adsorption Procedure and Mathematical Modeling

The captopril adsorption was performed in a batch system. The influence of magnetite was initially evaluated employing GO and GO·$Fe_3O_4$ to determine the effect of iron nanoparticles in GO surface on the adsorption efficiency. The adsorption equilibrium study was evaluated using 50 mg of GO·$Fe_3O_4$ and 50 mg $L^{-1}$ of initial concentration of captopril, pH 3.0, and different temperatures (20 °C, 30 °C, and 40 °C). The adsorption kinetic was performed employing different concentrations of CPT ≥ 98% (10–200 mg $L^{-1}$), 50 mg of adsorbent, at room temperature, and pH 3.0. Thermodynamic parameters were evaluated under the same conditions of the equilibrium study. Besides, this the influence of pH was verified in a pH range between 2.0 and 10.0, using HCl and NaOH solutions (0.1 mol $L^{-1}$) for adjustment of the pH of the solution. The effect of ionic strength was performed employing a gradual NaCl concentration (0.01–1 mol $L^{-1}$). Following this, the effect of initial concentration and adsorbent dosage were performed using 10–200 mg $L^{-1}$ of CPT and 12.5–50 mg of GO·$Fe_3O_4$, respectively. The adsorbent regeneration was carried out employing NaOH (0.25 mol $L^{-1}$) as the desorbing agent at room temperature for 1 h and 150 rpm. The residual concentration of CPT was measured using a UV-vis spectrophotometer (Shimadzu-1650PC) at λ = 217 nm. The removal percentage and adsorption capacity at equilibrium are shown in Table 1 [23].

Isotherms models were employed to understand the relationship between the adsorbate and adsorbent molecules. The Langmuir, Freundlich, and Sips models were used to fit the experimental results, and the equations are shown in Table 1 [24].

The kinetic study is useful to explain the mass transfer phenomenon and the effect of contact time on the adsorption rate. The models used in this work were Pseudo-first-order (PFO), Pseudo-second-order (PSO), and Elovich. Thermodynamic parameters were calculated by Van 't Hoff equation (Equation (10)) to determine the adsorption behavior [25,26]. The equilibrium and kinetic data were estimated through non-linear regression, using the Statistical 10 software (StatSoft, Tulsa, OK, USA). The coefficient of determination ($R^2$), adjusted coefficient of determination ($R^2_{adj}$), sum of squared errors (SSE), and average relative error (ARE) were used to verify the validity of models, and mean square error (MSE) are shown in Table 1 [27].

The mathematical modeling used to calculate the adjustment parameters are shown in Table 1.

**Table 1.** Equations of removal percentage, adsorption capacity, adsorption isotherms, kinetics, thermodynamics models, and mathematical modelling.

| Removal Percentage and Adsorption Capacity | Mathematical Expression | | Parameters |
|---|---|---|---|
| Removal percentage | $R\% = \dfrac{C_0 - C_e}{C_0} \times 100$ | (1) | $C_0$ is the initial concentration of CPT (mg L$^{-1}$); $C_e$ is the concentration of CPT at equilibrium (mg L$^{-1}$), $q_e$ is the adsorption capacity at equilibrium; $V$ is the volume (L); $m$ is the mass of adsorbent material (g) |
| Adsorption capacity | $q_e = \dfrac{(C_0 - C_e)\, V}{m}$ | (2) | |
| **Isotherm models** | **Mathematical expression** | | **Parameters** |
| Langmuir | $q_e = \dfrac{q_{max}\, K_L . C_e}{1 + K_L\, C_e}$ | (3) | $q_{max}$- maximum amount adsorbed (mg g$^{-1}$); $K_L$- is the Langmuir isotherm constant (L g$^{-1}$) |
| Freundlich | $q_e = K_F \cdot C_e^{1/n}$ | (4) | $K_F$- Freundlich constant ((mg g$^{-1}$) (L mg$^{-1}$)$^{-1/n}$); $n$- heterogeneity factor |
| Sips | $q_e = \dfrac{q_{Sips} K_{Sips} C_e^{\,n_{Sips}}}{1 + K_{Sips} C_e^{\,n_{Sips}}}$ | (5) | $K_{Sips}$- Sips isotherm constant (L mg$^{-1}$); $q_{Sips}$- maximum amount adsorbed (mg g$^{-1}$); $n_{Sips}$- heterogeneity factor |
| **Kinetic models** | **Mathematical expression** | | **Parameters** |
| Pseudo-first-order | $q_t = q_1\left(1 - e^{-k_1 \cdot t}\right)$ | (6) | $k_1$- pseudo-first-order kinetic constant (min$^{-1}$) |
| Pseudo-second-order | $q_t = \dfrac{t}{\left(\frac{1}{k_2 \cdot q_2^2}\right) + \left(\frac{t}{q_2}\right)}$ | (7) | $k_2$- pseudo-second-order kinetic constant (mg g$^{-1}$ min$^{-1}$) |
| Elovich | $q_e = \dfrac{1}{\beta}\, \ln(\alpha\beta t + 1)$ | (8) | $\alpha$- initial adsorption rate (mg g$^{-1}$ min$^{-1}$); $\beta$- desorption constant (g mg$^{-1}$) |
| **Thermodynamic** | **Mathematical expression** | | **Parameters** |
| Gibbs free energy variation ($\Delta G^0$) | $\Delta G^\circ = -RT\, lnK_e$ | (9) | $R$- gas constant (8.314 J mol$^{-1}$ K$^{-1}$); $T$- absolute temperature (K); $K_d$- thermodynamic equilibrium constant |
| van't Hoff | $lnK_e = \dfrac{\Delta S^\circ}{R} + \dfrac{\Delta H^\circ}{RT}$ | (10) | |
| Relationship of Gibbs free energy($\Delta G^0$), Enthalpy ($\Delta H^0$) and entropy ($\Delta S^0$) variation | $\Delta G^\circ = \Delta H^\circ - T\Delta S^\circ$ | (11) | |
| **Mathematical modeling** | **Mathematical expression** | | **Parameters** |
| Adjusted coefficient of determination ($R^2_{adj}$) | $R^2_{adj} = 1 - \left[\dfrac{(1 - R^2)(n - 1)}{n - k - 1}\right]$ | (12) | $q_{e,exp}$ and $q_{e,pred}$ are the adsorbed amounts obtained from the experiment and the isotherm and kinetic model, respectively; $n$ is the amount of data; $k$ is the number of parameters in the model |
| Sum squares error (SSE) | $SSE = \dfrac{1}{n}\sum\limits_{i=1}^{n}\left(q_{e,\,exp} - q_{e,\,pred}\right)^2$ | (13) | |
| Average relative error (ARE) | $ARE = \dfrac{100}{n}\sum\limits_{i=1}^{n}\left|\dfrac{q_{e,\,exp} - q_{e,\,pred}}{q_{e,\,exp}}\right|$ | (14) | |
| Mean square error (MSE) | $MSE = \dfrac{1}{n-P}\sum\limits_{i=1}^{n}\left(q_{e,exp} - q_{e,pred}\right)^2$ | (15) | $q_{e,exp}$ is the experimental value; $q_{e,pred}$ is the predicted value; $n$ is the number of experimental values; $p$ is the number of parameters according to the model. |

## 3. Results and Discussion

### 3.1. Fourier Transform Infrared Spectroscopy (FTIR)

The presence of functional groups was evaluated by FTIR analysis (Figure 1). According to the results, it was possible to observe an intense signal around 3405 cm$^{-1}$, corresponding to the stretching vibration of OH groups in GO composition [28]. The bands at 1734, 1651, 1342, and 1048 cm$^{-1}$ are related to the C=O stretch vibration, C=C bonds, C-OH vibration, and C-O bonds, respectively [29]. Besides this, for the GO·Fe$_3$O$_4$, it was

possible to verify the appearance of band around 617 cm$^{-1}$ characteristics of Fe-O, which suggests the presence of magnetite into GO surface [30,31].

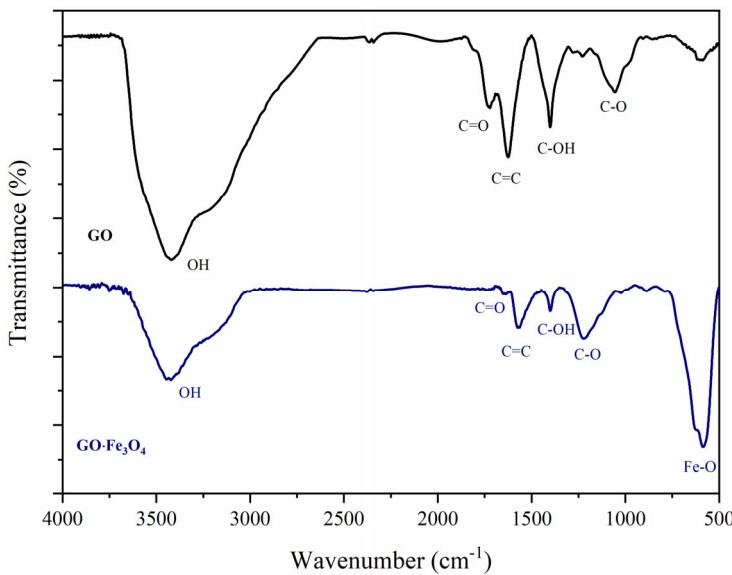

**Figure 1.** FTIR of GO and GO·Fe$_3$O$_4$.

### 3.2. X-ray Diffraction (XRD)

The XRD results of graphene oxide and magnetic graphene oxide are shown in Figure 2. The GO diffraction shows the characteristic signal around $2\theta \approx 10.5°$, corresponding to the (001) plane, and the absence of a peak at $2\theta \approx 26°$ indicate the complete oxidation of graphite (precursor material) [32]. The peak at $2\theta \approx 44.2°$ (100) suggests a small distance between GO layers. For the GO with iron oxide nanoparticles, it was possible to verify the appearance of peaks around $2\theta \approx 30.3°$, $35.2°$, $43.5°$, $57.02°$, and $63.3°$, corresponding to the indices (220), (311), (400), (511) and (440), typical of magnetite [33]. The position and intensity of all peaks are compatible with Fe$_3$O$_4$ (ICDD card no:88-0315). The absence of GO signal in the GO·Fe$_3$O$_4$ diffractogram is attributed to the surface coated by iron nanoparticles, which is possible to observe in SEM results (Figure 3) [21].

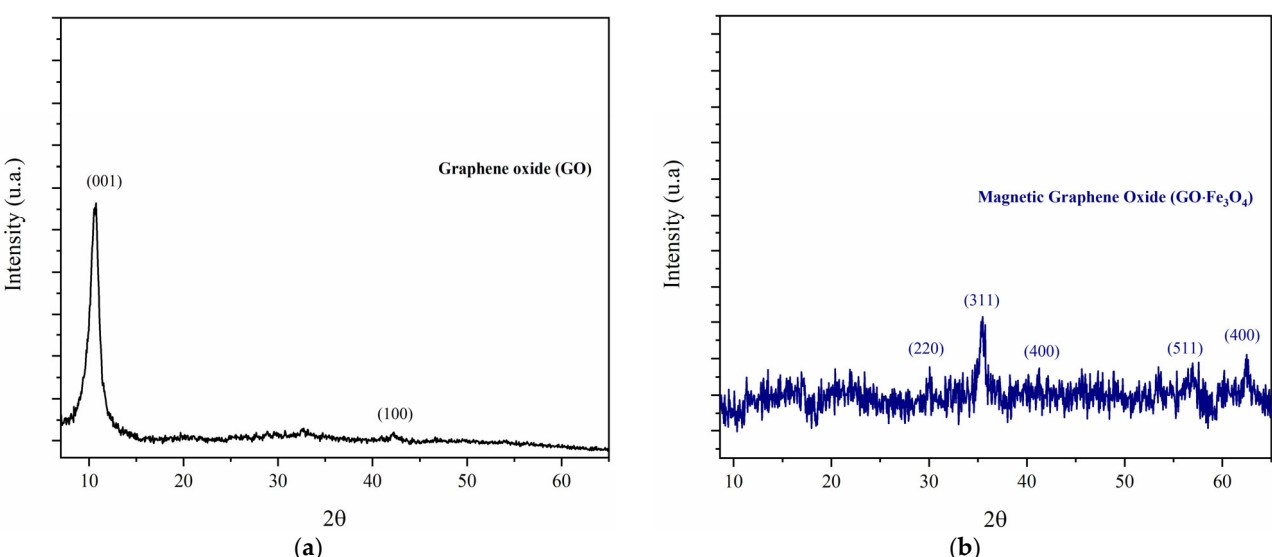

**Figure 2.** XRD of (**a**) GO and (**b**) GO·Fe$_3$O$_4$.

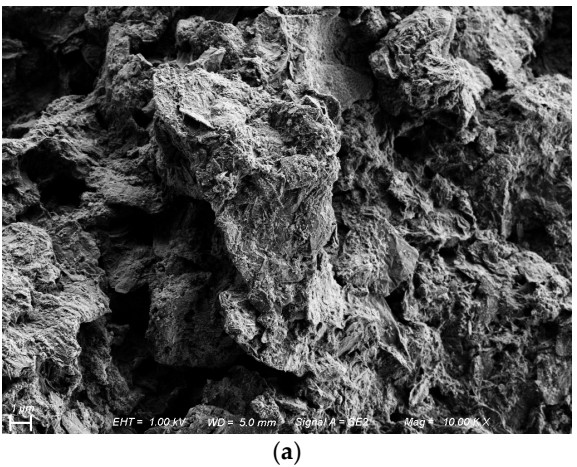

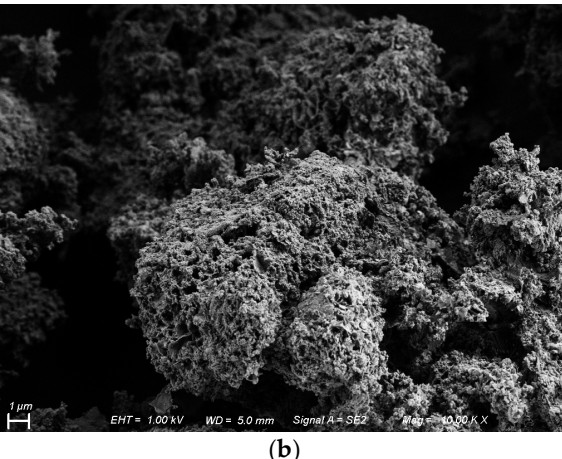

(**a**)    (**b**)

**Figure 3.** SEM of (**a**) GO and (**b**) GO·$Fe_3O_4$.

### 3.3. Scanning Electron Microscopy (SEM)

The morphology of GO and GO·$Fe_3O_4$ was determined by scanning electron microscopy (SEM) and is shown in Figure 3. For the GO (Figure 3a), it was possible to verify a wrinkles sheet, which is characteristic of a nanomaterial with few layers [34]. In Figure 3b, for the magnetic graphene oxide, the GO surface was covered by magnetite. The high amount of $FeCl_2$ employed in the magnetization process results in a magnetic nanocomposite with high magnetic nanoparticle content [21].

### 3.4. Raman Spectroscopy

The Raman spectroscopy results are shown in Figure 4. The three bands can be observed for GO andgraphene oxide-based magnetic nanoadsorbent (GO·$Fe_3O_4$). For both materials, it was possibile to observe the bands at 1366, 1580, and 2698 cm$^{-1}$ that represent the bands D (defects and disorder of graphite structure), G (sp$^2$ vibration by carbon atoms), and 2D (second order of the D band), which is characteristic of graphene materials [21]. Additionally, the $I_D/I_G$ ratio increased after the magnetization procedure, showing values of 0.96 and 1.22 for GO and GO·$Fe_3O_4$. The increases are related to the involvement of carbon atoms in partial $Fe^{3+}$ reduction [35].

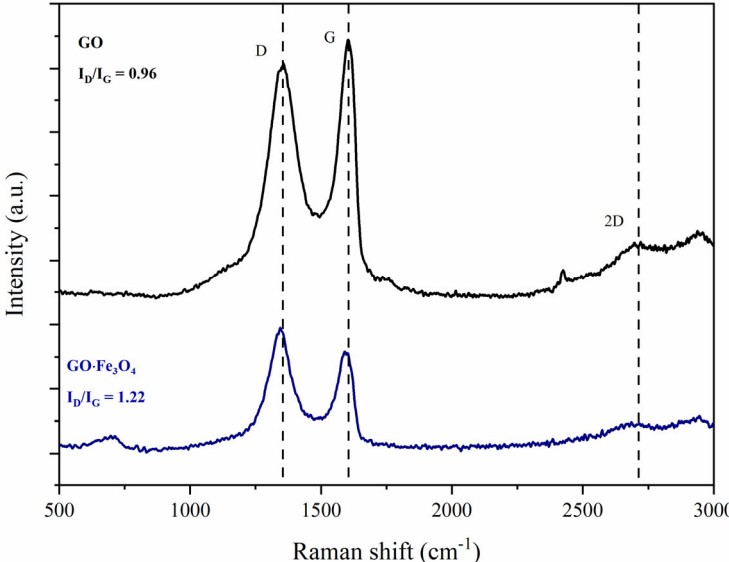

**Figure 4.** Raman spectroscopy of GO and GO·$Fe_3O_4$.

Nonetheless, the co-precipitation method is widely used to synthesize magnetic nanoparticles, due to the low energy requirements, low temperature, and high-quality products. In this work, the magnetization process increased the $I_D/I_G$ ratio; however, the low values showed that the MGO was not significantly affected by the procedure. Alongside this, using ammonium hydroxide as a precipitant agent, the graphene structure was preserved, resulting in excellent magnetic graphene oxide surface and low values for $I_D/I_G$ ratio [21]. Zhang et al. [36] synthesized a magnetic nanocomposite, employing a microwave absorbing method with a mixture of $Fe^{2+}$ and $Fe^{3+}$ (1:2 molar ratio). The results of $I_D/I_G$ ratio showed a high disorder in the nanocomposite structure. Nevertheless, Hatel et al. [37] developed $GO/Fe_3O_4$ nanorods using ultrasonic radiation and NaOH as the precipitating agent. Raman spectroscopy results demonstrated a nanocomposite with a low crystallinity degree when compared to the GO sample.

### 3.5. Vibrating Sample Magnetometer (VSM)

Figure 5 shows the magnetization profile of magnetic graphene oxide at room temperature. According to the graph, it was possible to verify that $GO \cdot Fe_3O_4$ presented a ferromagnetic behavior; magnetization saturation ($M_S$), remanent magnetization ($M_R$), and $M_R/M_S$ ratio values were 45 emu $g^{-1}$, 3.44 emu $g^{-1}$, and 0.076, respectively. Similarly, Ghosh et al. [38] developed a magnetic GO functionalized with nitrogen groups, and the materials showed a ferromagnetic behavior.

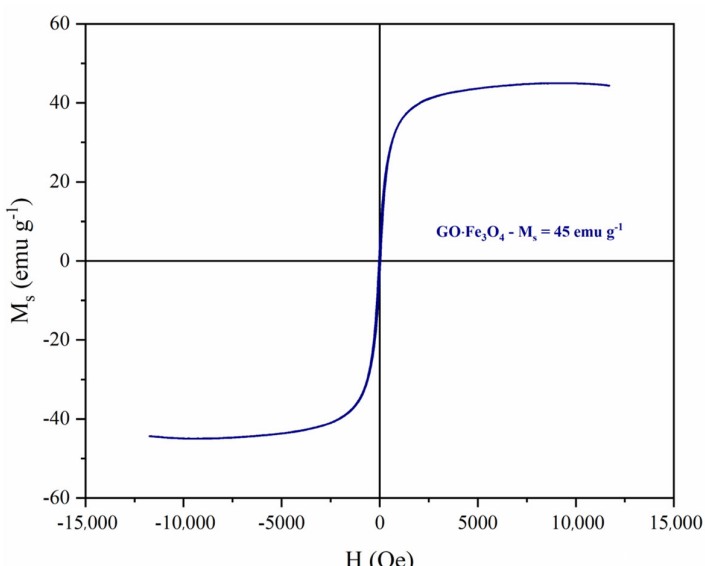

**Figure 5.** VSM of $GO \cdot Fe_3O_4$.

### 3.6. Captopril Adsorption

3.6.1. Effect of Magnetite Incorporation onto Graphene Oxide Surface

The synthesis of nanocomposites is an important tool for the design of adsorbent materials with different characteristics and properties. In this study, the effect of the magnetite incorporation on the GO surface on the captopril adsorption behavior was evaluated (Figure 6).

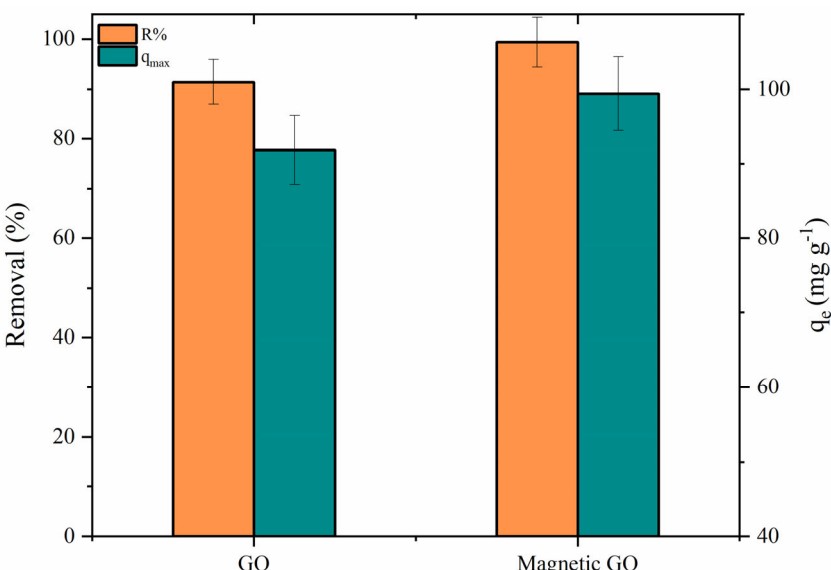

**Figure 6.** Effect of magnetite incorporation onto the graphene oxide surface. Adsorbent dosage (0.5 g L$^{-1}$), initial concentration of CPT (50 mg L$^{-1}$), pH 3.0, and 293.15 K.

According to Figure 6, it was possible to verify that both adsorbents exhibited excellent performance when removing the emerging pollutant. However, the graphene oxide-based magnetic adsorbent presented the highest adsorption capacity and removal percentage values compared to the pristine carbon nanomaterial. The GO and GO·Fe$_3$O$_4$ display removal values of 91.42 and 99.43% and adsorption capacities of 91.84 and 99.41 mg g$^{-1}$, respectively. The development of composite materials not only improve the physicochemical stability, but also can enhance their efficiency as an adsorbent. Recently, diverse studies have reported the effects of magnetite on the GO surface in terms of the adsorption performance [21,39,40].

### 3.6.2. Effect of Initial Concentration of CPT and Adsorbent Dosage

The initial adsorbate concentration and the adsorbent dosage play an important role in the adsorption efficiency, due to the higher ability to overcome the mass transfer resistance phenomenon and the more available surface area, respectively. Here, the effect of the initial concentration of CPT and GO·Fe$_3$O$_4$ was investigated in the range of 10–200 mg L$^{-1}$ and 0.125–1.0 g L$^{-1}$, respectively.

The influence of CPT concentration is shown in Figure 7. Remarkably, the adsorption capacity values improved with the gradual increase in the initial concentration of CPT (e.g., the q$_e$ value raised from 14.68 to 390.31 mg g$^{-1}$ when the CPT concentration was changed from 10 to 200 mg L$^{-1}$), thus implying that CPT adsorption is favored at higher concentrations. However, the removal percentage displayed a slight decrease with rising concentration. The removal value reduced by around 23% from the lowest to the highest concentration. The improvement in the adsorption performance can be attributed to the higher concentration gradient, impelling the motions of the adsorbate to the boundary layer, favoring the interaction between the liquid phase and the solid phase (surface of the adsorbent) and, therefore, the mass transfer phenomenon [41,42].

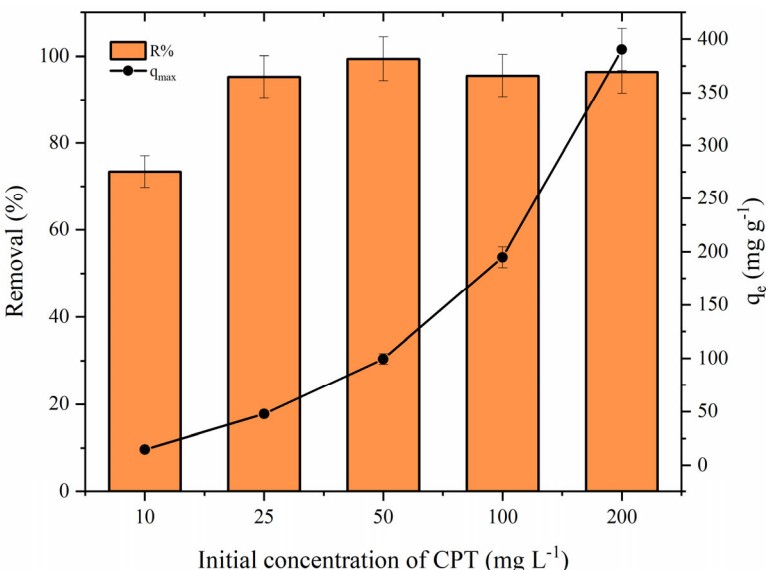

**Figure 7.** Effect of the initial concentration of CPT (GO·Fe$_3$O$_4$ dosage (0.5 g L$^{-1}$), C$_0$ = 10−200 mg L$^{-1}$, pH 3.0, and 293.15 K).

Regarding the effect of adsorbent dosage (Figure 8), it was observed that an initial increase in the GO·Fe$_3$O$_4$ mass lead to an improvement in the removal percentage values. However, at the highest dosages (0.75 and 1.0 g L$^{-1}$), a slight reduction was verified (the removal value decreased from 99.43 to 88.71% when the adsorbent dosage changed from 0.5 to 1.0 g L$^{-1}$). On the other hand, the adsorption capacity decreased inversely with the gradual rise in the adsorbent quantity. This is related to the imbalance between available adsorption sites and absorbate molecules, which is caused bt higher adsorbent dosages [43,44].

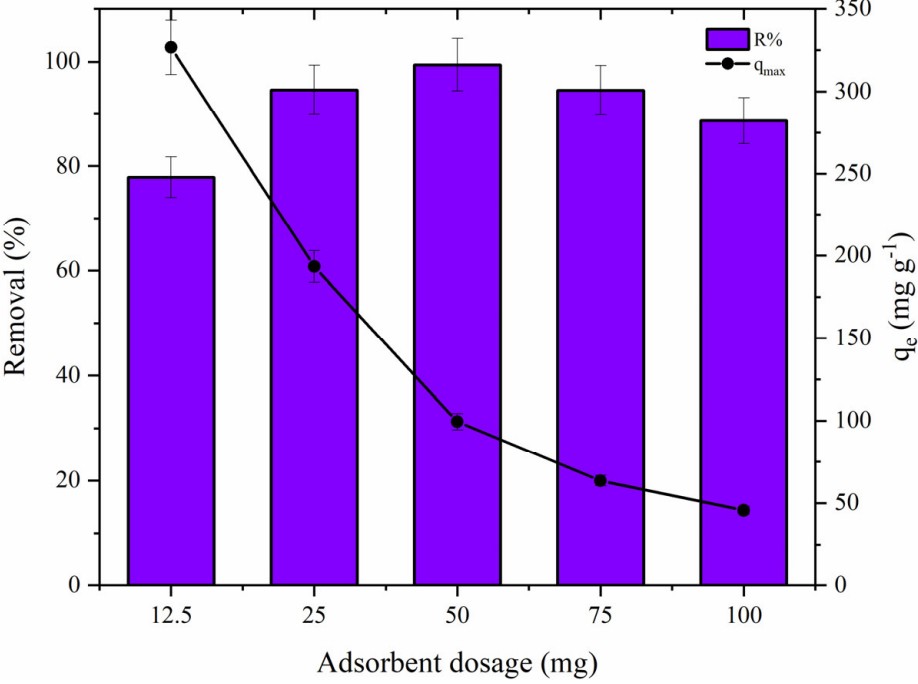

**Figure 8.** Effect of the adsorbent dosage on CPT adsorption (GO·Fe$_3$O$_4$ dosage (0.125–1.0 g L$^{-1}$), initial concentration of CPT (50 mg L$^{-1}$), pH 3.0, and 293.15 K).

### 3.6.3. Effect of pH and Adsorption Mechanisms

The pH of the solution is an important experimental parameter, due to its influence on chemical speciation and adsorbent surface charge. The effect of pH on CPT adsorption was evaluated in the range of 2–9, keeping other parameters constant. At the same time, the zero potential of charge ($pH_{ZPC}$) was estimated by the typical 11-point experiment.

From the results presented in Figure 9, it was possible to observe that the pH had a significant effect on the adsorption capacity and removal percentage values. The adsorption was reached at pH 3.0 ($q_e$ = 99.41 mg g$^{-1}$ and R% = 99.43). Captopril is a drug used for arterial hypertension that contains an ionizable group (-COOH, $pK_a$ = 3.7), i.e., it has an anionic character in this pH condition [45]. Under acidic conditions, the GO·Fe$_3$O$_4$ presents a positive surface charge ($pH_{ZPC}$ = 7.41), which allows the occurrence of electrostatic interactions between the adsorbent and adsorbate (pH 3.0). In contrast, in the range between 4–7, the adsorbent and captopril have a cationic characteristic, decreasing the affinity of adsorbate by binding sites. At the same time, in acidic media, the H$^+$ ions can compete for the available active sites, justifying the considerable reduction in the adsorption performance. In addition, under alkaline conditions, the adsorbent material exhibits an anionic surface, also affecting the interaction between the systems [21].

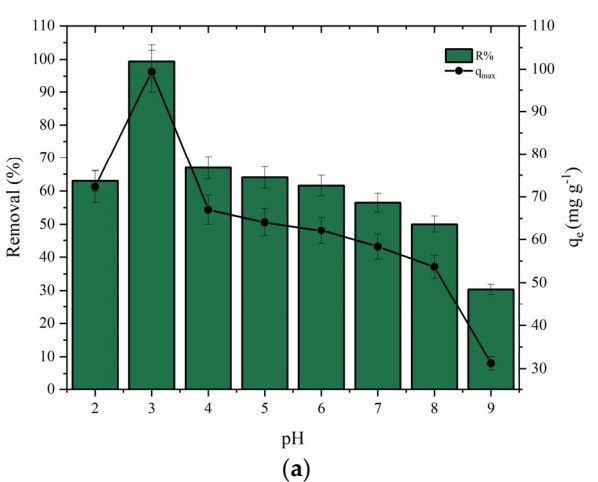

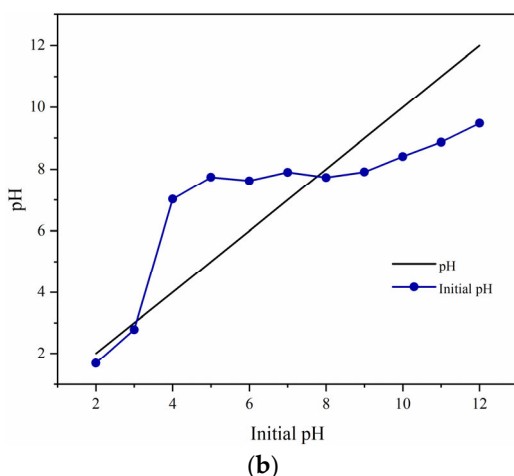

(a)  (b)

**Figure 9.** (**a**) Effect of pH on CPT adsorption (C$_0$ = 50 mg L$^{-1}$, pH = 2–9, adsorbent dosage = 0.5 g L$^{-1}$, V = 100 mL, and 293.15 K) and (**b**) Zero point of charge of GO·Fe$_3$O$_4$.

Adsorption mechanisms are closely related to the chemical structure of the adsorbate, such as functional groups, aromaticity, hydrophilicity, and pH sensitivity (ionizable groups). Additionally, the interaction depends on the charge and specific surface area of the adsorbent material. Thus, some hypotheses regarding the CPT adsorption onto GO·Fe$_3$O$_4$ were proposed (Figure 10). Considering the influence of pH solution on the chemical speciation of adsorbate and the adsorbent surface charge, this suggests that the captopril adsorption phenomenon is mainly governed by electrostatic interactions (optimal condition was reached at pH 3.0) [45,46]. Although attraction forces can act on the CPT removal, the abundance of oxygenated functional groups in nanoadsorbent structure and carboxylic acids on the captopril molecule also allows the occurrence of other interactions, such as hydrogen bonds (Yoshida H-bonding and dipole-dipole hydrogen bonding) [44,47].

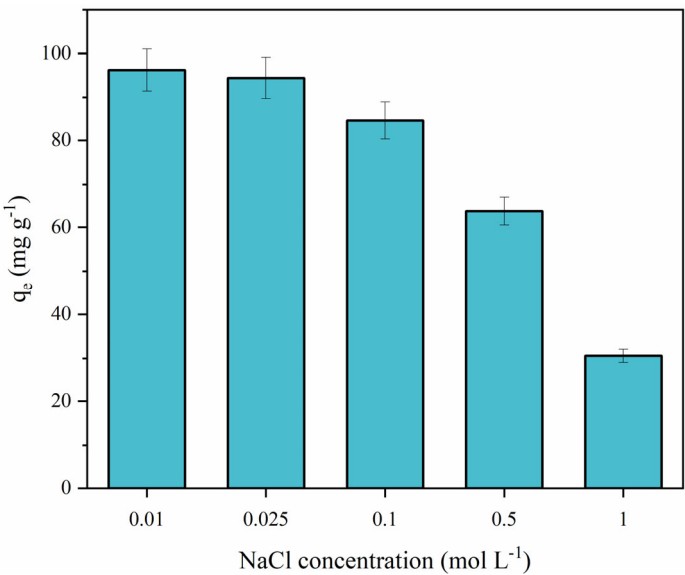

**Figure 10.** Hypotheses of the captopril adsorption mechanism.

### 3.6.4. Effect of Ionic Strength

The evaluation of the effect from diverse experimental conditions is an important requirement considering the potential application in wastewater treatment systems. In this study, the influence of ionic strength on CPT adsorption was performed using different concentrations of sodium chloride (0.01–1.0 mol L$^{-1}$), keeping other parameters constant. As shown in Figure 11, it was possible to observe that an increase in NaCl molarity caused a considerable reduction in the adsorption capacity value. For instance, the system without NaCl presents a $q_e$ of 99.41 mg g$^{-1}$, while, at a concentration of 1.0 mol L$^{-1}$, the adsorption capacity was 32.38 mg g$^{-1}$. The Na$^+$ and Cl$^-$ ions can hamper the binding between the adsorbate and adsorbent due to the competition by adsorption sites. Additionally, the presence of inorganic salts may decrease the specific surface area, as well as reduce the solubility of the adsorbate [44,48,49].

**Figure 11.** Effect of ionic strenght on CPT adsorption (C$_0$ = 50 mg L$^{-1}$, pH = 3.0, adsorbent dosage = 0.5 g L$^{-1}$, NaCl concentration (0.01–1.0 mol L$^{-1}$), V = 100 mL, and 293.15 K).

### 3.6.5. Kinetic Modeling

Kinetic models are valuable mathematical expressions, widely used to understand the adsorption behavior in relation to the experimental time. In this work, three models were

employed: pseudo-first-model, pseudo-second-order, and Elovich to construct the kinetic profile and calculate the adjustment parameters for captopril adsorption onto $GO \cdot Fe_3O_4$. The kinetic study was conducted at room temperature using different concentrations of adsorbate. Table 2 shows the adjust of experimental data to nonlinear kinetic models.

**Table 2.** Kinetic parameters for captopril adsorption onto $GO \cdot Fe_3O_4$.

| Concentration (mg L$^{-1}$) | 10 | 25 | 50 | 100 | 200 |
|---|---|---|---|---|---|
| Pseudo-first-order model (PFO) | | | | | |
| $q_1$ (mg g$^{-1}$) | $13.44 \pm 0.34$ [a] | $45.69 \pm 0.97$ | $96.06 \pm 0.91$ | $186.32 \pm 1.81$ | $386.58 \pm 0.60$ |
| $k_1$ (min$^{-1}$) | $0.375 \pm 0.11$ | $0.205 \pm 0.04$ | $0.606 \pm 0.15$ | $0.403 \pm 0.05$ | $0.804 \pm 0.05$ |
| $R^2$ | 0.948 | 0.966 | 0992 | 0.991 | 0.998 |
| $R^2_{adj}$ | 0.935 | 0.957 | 0.990 | 0.988 | 0.997 |
| ARE (%) | 5.67 | 4.28 | 2.09 | 1.96 | 0.25 |
| SSE | 0.84 | 6.17 | 5.77 | 6.47 | 2.51 |
| MSE (mg g$^{-1}$)$^2$ | 1.33 | 9.69 | 9.06 | 38.17 | 3.94 |
| Pseudo-second-order model (PSO) | | | | | |
| $q_2$ (mg g$^{-1}$) | $13.59 \pm 0.35$ | $47.99 \pm 0.62$ | $97.03 \pm 0.87$ | $190.11 \pm 1.21$ | $387.53 \pm 0.48$ |
| $k_2$ (g mg$^{-1}$ min$^{-1}$) | $0.049 \pm 0.02$ | $0.007 \pm 0.001$ | $0.025 \pm 0.01$ | $0.005 \pm 0.001$ | $0.023 \pm 0.004$ |
| $R^2$ | 0.966 | 0.991 | 0.994 | 0.997 | 0.983 |
| $R^2_{adj}$ | 0.957 | 0.988 | 0.992 | 0.996 | 0.978 |
| ARE (%) | 5.04 | 1.97 | 1.82 | 1.11 | 2.59 |
| SSE | 0.53 | 1.51 | 3.93 | 7.16 | 16.35 |
| MSE (mg g$^{-1}$)$^2$ | 0.83 | 3.36 | 6.17 | 11.25 | 9.98 |
| Elovich model | | | | | |
| $\alpha$ (mg g$^{-1}$ min$^{-1}$) | $10.48 \pm 0.21$ | $15.35 \pm 0.59$ | $16.62 \pm 1.02$ | $4.57 \pm 0.97$ | $3.95 \pm 1.02$ |
| $\beta$ (g mg$^{-1}$) | $0.997 \pm 0.01$ | $0.222 \pm 0.05$ | $0.273 \pm 0.026$ | $0.118 \pm 0.02$ | $0.066 \pm 0.002$ |
| $R^2$ | 0.995 | 0.994 | 0.998 | 0.999 | 0.999 |
| $R^2_{adj}$ | 0.993 | 0.992 | 0.997 | 0.998 | 0.998 |
| ARE (%) | 2.53 | 2.88 | 1.02 | 0.62 | 0.19 |
| SSE | 0.14 | 2.14 | 2.41 | 1.81 | 1.20 |
| MSE (mg g$^{-1}$)$^2$ | 0.23 | 2.27 | 4.01 | 2.84 | 2.57 |

[a] (Mean $\pm$ standard deviation).

According to the results presented in Table 2, it was possible to assume that the captopril adsorption onto graphene oxide-based magnetic nanoadsorbent was well-described by the Elovich model in all concentrations tested. This because the model exhibited high values for coefficient of determination ($R^2 \geq 0.994$) and low values for error functions (ARE, MSE and SSE). Additionally, the Elovich parameters related to sorption rate ($\alpha$) and surface coverage ($\beta$), respectively, indicated that the adsorption occurs in the heterogeneous surface, i.e., the adsorbent exhibits adsorption sites with different energy levels [24,50].

The kinetic profile of captopril adsorption is shown in Figure 12. As demonstrated by kinetic curves, it was possible to verify that the adsorption presents a characteristic behavior of fast kinetic, with three different stages. In the first minutes, a high amount of adsorbate is removed, followed by a stationary phase (plateau) and, finally, equilibrium is achieved. In addition, the adsorption capacity at equilibrium is proportional to the initial concentration of the contaminant. The increase in adsorption efficiency can be attributed to greater ease when overcoming of the mass transfer phenomenon and effective occupation of binding sites [51].

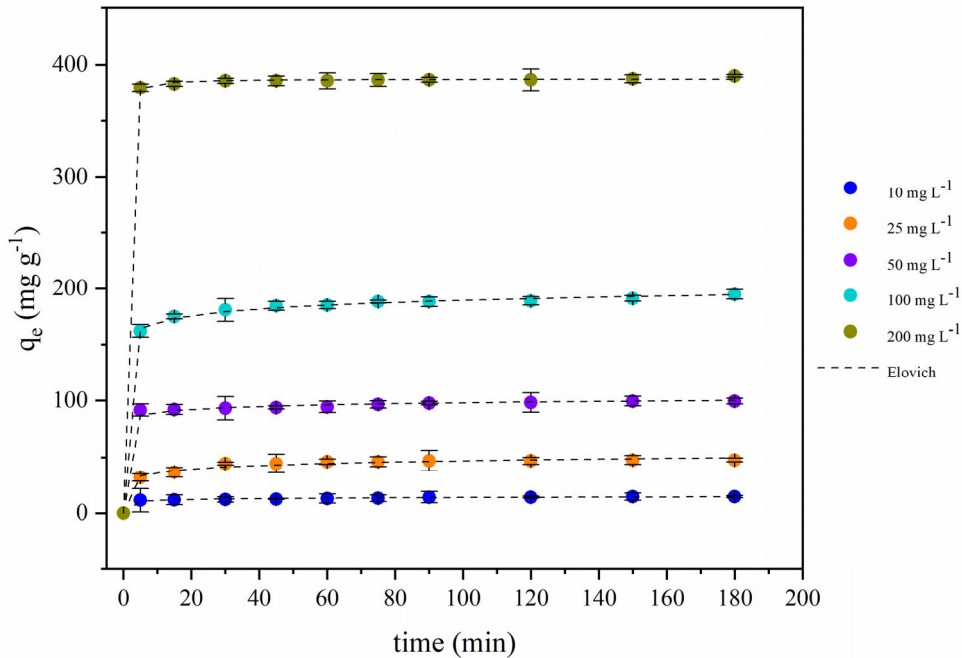

**Figure 12.** Kinetic curves for CPT adsorption onto GO·Fe$_3$O$_4$.

3.6.6. Adsorption Equilibrium Isotherms and Thermodynamic Study

The isotherm models are useful mathematical expressions that help us to understand the interaction between adsorbate and adsorbent; they are also used to verify the influence of temperature in the adsorption equilibrium. In this present work, Langmuir, Freundlich, and Sips isotherm models were used to predict the best adjustment of equilibrium data at different temperatures. As shown in Table 3, it was possible to determine that the Sips isotherm was the best model to describe the CPT adsorption onto GO·Fe$_3$O$_4$. Concerning values for coefficients of determination ($R^2$) and error functions (ARE and SSE), the descending order of fit of the models is as follows:

<p align="center">Sips < Freundlich < Langmuir</p>

The Sips model is a combined equation of the Langmuir and Freundlich isotherms, used to predict the adsorption on the heterogenous surface [52]. Regarding the Sips constant associated to the heterogeneity (ns), it is remarkable that the heterogeneity of the system increased with the temperature (ns = 2.26 at 20 °C and ns = 7.88 at 40 °C). Additionally, the maximum adsorption capacity estimated by this model decreased with the rise in the temperature, indicating that adsorption is an exothermic process (in agreement with the thermodynamic study) [45]. On the other hand, in the Sips constant related to equilibrium (Ks), a considerable decrease was observed with the gradual increase in temperature, i.e., suggesting that the process is disadvantaged at high temperatures [53].

Figure 13 shows the Sips isotherms curves. According to the results, it is possible to observe a characteristic "L-shape" type for 20 and 30 °C, indicating an adsorbent surface with a high presence of binding sites [54]. However, for 40 °C, the isotherm showed an "S-shape" suggesting a solute–solute attraction and/or an adsorption site competition [55].

**Table 3.** Equilibrium parameters for CPT adsorption onto GO·Fe$_3$O$_4$.

| Temperature | 20 °C | 30 °C | 40 °C |
|---|---|---|---|
| *Langmuir* | | | |
| $q_{max}$ (mg g$^{-1}$) | 101.93 ± 1.76 [a] | 101.49 ± 3.35 | 99.28 ± 5.96 |
| $K_L$ (L mg$^{-1}$) | 0.771 ± 0.46 | 0.43 ± 0.055 | 0.057 ± 0.97 |
| $R^2$ | 0.979 | 0.953 | 0.979 |
| $R^2_{adj}$ | 0.973 | 0.941 | 0.973 |
| ARE (%) | 3.82 | 5.69 | 4.92 |
| SSE | 15.71 | 26.73 | 8.54 |
| MSE (mg g$^{-1}$)$^2$ | 16.38 | 17.42 | 19.55 |
| *Freundlich* | | | |
| $K_F$ ((mg g$^{-1}$) (L$^{-1}$)$^{-1/n}$ | 74.51 ± 2.21 | 5.01 ± 3.32 | 2.61 ± 1.97 |
| $n$ | 10.95 ± 3.29 | 45.85 ± 2.05 | 16.54 ± 1.44 |
| $R^2$ | 0.986 | 0.997 | 0.997 |
| $R^2_{adj}$ | 0.982 | 0.995 | 0.997 |
| ARE (%) | 3.84 | 1.99 | 1.01 |
| SSE | 10.21 | 2.64 | 1.87 |
| MSE (mg g$^{-1}$)$^2$ | 9.77 | 5.67 | 15.43 |
| *Sips* | | | |
| $q_s$ (mg g$^{-1}$) | 100.41 ± 0.97 | 90.47 ± 1.19 | 70.72 ± 2.56 |
| $K_s$ (L mg$^{-1}$) | 0.444 ± 0.14 | 0.255 ± 0.65 | 0.082 ± 0.001 |
| $ns$ | 2.26 ± 1.15 | 3.91 ± 1.74 | 7.88 ± 0.68 |
| $R^2$ | 0.998 | 0.999 | 0.999 |
| $R^2_{adj}$ | 0.997 | 0.998 | 0.999 |
| ARE (%) | 1.24 | 1.84 | 0.09 |
| SSE | 1.85 | 0.55 | 0.01 |
| MSE (mg g$^{-1}$)$^2$ | 3.41 | 2.48 | 3.01 |

[a] (Mean ± standard deviation).

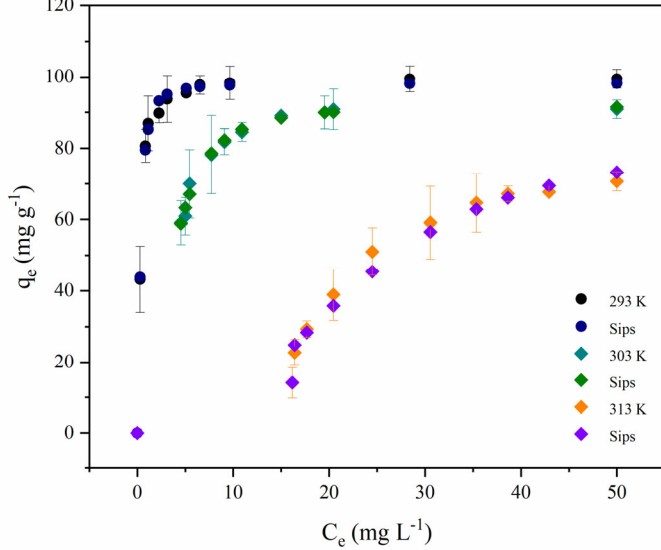

**Figure 13.** Sips model for CPT adsorption onto GO·Fe$_3$O$_4$.

To understand adsorption behavior under the influence of temperature, this study was performed at different conditions (293.15, 313.15, and 333.15 K). The thermodynamic equilibrium constant ($K_e$) was calculated using the $K_{Sips}$ (best adjustment for isotherm model), and the captopril molecular weight (217.29 g mol$^{-1}$) [16,56–59] Thermodynamic parameters for CPT adsorption onto GO·Fe$_3$O$_4$ were calculated by the Van 't Hoff equation and are shown in Table 4. From the values for Gibbs free energy variation ($\Delta G^0$), it was

possible to assume that the captopril adsorption was a thermodynamically favorable process [56]. Furthermore, the negative value for enthalpy change ($\Delta H^0$) indicated exothermic adsorption [47]. Additionally, the magnitude ($-64.19$ kJ mol$^{-1}$) suggests that the adsorption phenomenon is mainly governed by chemical interactions [60]. Regarding the entropy change ($\Delta S^0$), the negative value is related to a decrease in the randomness of the system.

**Table 4.** Thermodynamic parameters for CPT adsorption onto $GO \cdot Fe_3O_4$.

| T(K) | $K_e$ | $\Delta G^0$ (kJ mol$^{-1}$) | $\Delta H^0$ (kJ mol$^{-1}$) | $\Delta S^0$ (kJ mol$^{-1}$ K$^{-1}$) |
|---|---|---|---|---|
| 293.15 | 96476.76 | $-27.92$ | | |
| 303.15 | 55408.95 | $-27.52$ | $-64.19$ | $-0.12$ |
| 313.15 | 17817.78 | $-25.48$ | | |

The adsorption performance was compared to previous studies employing different adsorbents and experimental conditions. It is possible to observe, in Table 5, that all the materials employed in CPT removal showed a removal percentage. Karperiski et al. [16] synthesized an activated carbon with $ZnCl_2$ from Caesalpina ferrea (CFAC) and obtained a $q_{max}$ of 535.5 mg g$^{-1}$. The results showed that the best adsorbent performance was reached by CFCAC.1.5 and the process was spontaneous and favorable for removal of 97.67% of CPT from the aqueous solution. Cunha et al. [9] developed an activated carbon derivate from Butia catarinensis (ABc) with a high surface area (1267 m$^2$ g$^{-1}$) and with high efficiency removal (up to 99%). The maximum adsorption capacity was dependent on the temperature, rising with the temperature. In this work, magnetic graphene oxide removed up to 99.0% of the drug, with a high adsorption capacity (100.41 mg g$^{-1}$) at pH 3.0. Therefore, the magnetite functionalization improved the adsorption process, allowing easy adsorbent regeneration and reusability and avoiding centrifugation/filtration steps.

**Table 5.** Removal percentage of different adsorbents of CPT.

| Adsorbents | pH | Removal (%) | Reference |
|---|---|---|---|
| Activated carbon functionalized with $ZnCl_2$ | 7.0 | 97.67 | [16] |
| Activated carbon derivate from Butia catarinensis | 7.0 | <99.0 | [9] |
| Microporous carbons | Not informed | 97.0 | [61] |
| Magnetic graphene oxide | 3.0 | <99 | This work |

3.6.7. Regeneration and Reuse

The study of the regeneration of adsorbents is a crucial step in the adsorption processes, aiming at the development and application of materials with high performance and cost-effectiveness. In this work, the regeneration of $GO \cdot Fe_3O_4$ was performed using a solution of sodium hydroxide (0.25 mol L$^{-1}$), stirring at room temperature for 60 min. Thus, five consecutive adsorption-desorption cycles were conducted to estimate the efficiency for several experiments. According to Figure 14, it was possible to observe that the nanoadsorbent exhibits high efficiency, even after diverse cycles of regeneration and reuse. The percentage removal value displayed a slight decrease over time, which can be attributed to incomplete desorption and mass loss during the adsorbent recovery step [26,47].

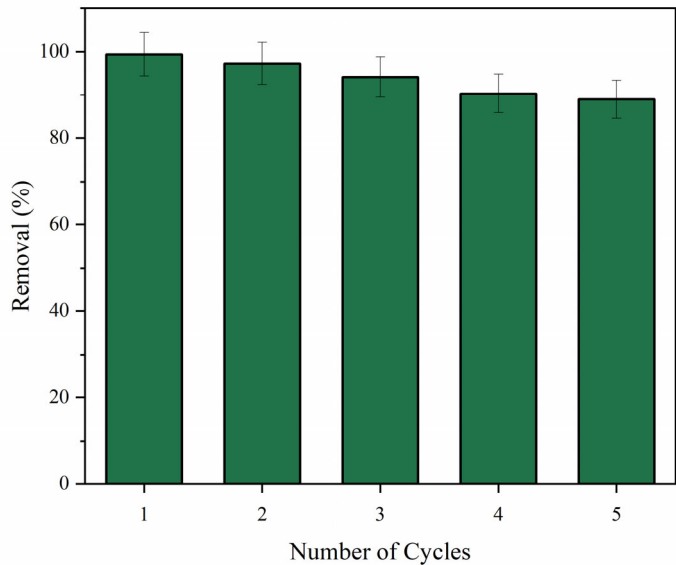

**Figure 14.** GO·Fe$_3$O$_4$ performance after several adsorption cycles.

## 4. Conclusions

The magnetic graphene oxide was synthesized employing a straightforward method with low energy requirements and easy operation. The FTIR, XRD, Raman, SEM, and VSM techniques demonstrated that the nanocomposite was successfully obtained, exhibiting excellent properties (crystalline structure with low defects) and high value for saturation magnetization (Ms = 45 emu g$^{-1}$). The graphene oxide-based magnetic nanoadsorbent exhibits higher performance when compared with the pristine carbon nanomaterial. The maximum adsorption capacity and removal percentage (99.43% and 100.41 mg g$^{-1}$) was reached at pH 3.0, using 50 mg of adsorbent, 50 mg L$^{-1}$ of CPT, and 293.15 K. These results can be attributed to the synergic effect between graphene oxide and iron oxide nanoparticles. The pH dependence and influence of the adsorbent type suggested that the CPT adsorption onto GO·Fe$_3$O$_4$ is mainly driven by electrostatic interactions and hydrogen bonds. Sips and Elovich models were well-suited to describe the equilibrium and adsorption kinetics data, assuming a process on a heterogeneous surface. The thermodynamic study inferred that the adsorption was spontaneous, exothermic, and occurred predominantly by chemical mechanisms. Additionally, the GO·Fe$_3$O$_4$ demonstrated high efficiency after five adsorption/desorption cycles. From the experimental results, it can be concluded that the magnetic adsorbent is a promising material for wastewater management, especially regarding emerging pollutants like captopril.

**Author Contributions:** M.P.d.O., C.S.: Methodology, Investigation. T.d.R.S., F.d.S.B., L.B.: Investigation, Writing—Original Draft. E.I.M., W.J.d.S.G.: Visualization, Data Curation. A.H.d.O., L.F.O.S.: Visualization, Writing—Review & Editing. C.R.B.R.: Conceptualization, Writing—Review & Editing, Supervision (cristianorbr@gmail.com). All authors have read and agreed to the published version of the manuscript.

**Funding:** This research received no external funding.

**Data Availability Statement:** Not applicable.

**Acknowledgments:** The authors thank Laboratório de Materiais Magnéticos Nanoestruturados—LaMMaN, Laboratório de Magnetismo e Materiais Magnéticos—LMMM, UFSM, and CAPES, CNPq, FAPERGS (TO 22/2551-0000609-4) for their support.

**Conflicts of Interest:** The authors declare no conflict of interest.

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
