# Peer review of "Efficient Uptake of Angiotensin-Converting Enzyme II Inhibitor Employing Graphene Oxide-Based Magnetic Nanoadsorbents"

_water, doi:10.3390/w15020293_

Round 1
Reviewer 1 Report
Manuscript 2128177 Water.
The paper describes the application of magnetic graphene oxide for water remediation. The subject of the paper is interesting. The materials characterization is broad. However, section of the Discussion contains some apodictic statements and some errors. This paper is suitable for the journal, but some parts should be improved.
MAJOR REVISION should follow the recommendations below.
A few specific comments:
p.4, Table 1: In the table at least two equations (9) and (10) must be corrected by the authors. Please check all the mathematical expressions.
p.4, Table 1: the correct value of the universal gas constant R is 8.314 J mol-1 K-1
p.6, fig.2(b): Why is the GO peak (001) completely missing?
p.8, line 208: “Fig.5” should be “Fig.6”.
p.9, line 214: Please, give an example where the GO-Fe3O4 composite has better physicochemical stability.
p.9, line 228: “The removal value reduces by around 23% from the lowest to the highest concentration” the sentence should be “The removal value increases by around 23% from the lowest to the highest concentration”
p.11, fig.10: In the structural formula of CPT a CH3 is missing. Consider that the formula is C9H15NO3S.
p.12, line 309: It is important that the Elovich equation is an empirical function. Hence, the assignment of physical meaning to the value of its parameters alpha and beta is questionable.
p.15 Table 4: Thermodynamic parameters are not consistent. Using the equation DG°=DH°-TDS° we get -6,2 kJ mol-1.
p. 15, Conclusions: In the conclusions, it is important to point out that the pH=3 conditions used in this experimental study are not reasonably applicable in a sewage treatment plant.
MINOR REVISION:
p.1, line 19: “captopril” should be “captopril (CPT)”
p.2, line79: “300 mL of distilled water…” the phrase is suspended (incomplete).
p.4, line at the bottom: “Sun squares error” should be “Sum squares error”
p.15, line371: “Accordin” should be “According”
Fig. 2(b): Why is the graphene oxide peak (001) totally absent?
p.4 line 208: “According to Fig. 5,” should be “According to Fig. 6”.
Author Response
We thank you for the opportunity to submit our revised paper to the Water. We revised the manuscript with great attention. The Referee's comments provide us with detailed and very useful reports. The modifications are highlighted in yellow color in the final version of the manuscript. By addressing their comments in the revised version, we are confident that the paper has been considerably improved, thus shaping into a publishable form. Please, find below the answers to the referee's comments.
With kind regards
Prof. Dr. Cristiano Rodrigo Bohn Rhoden
Corresponding Author
Reviewer 1
The paper describes the application of magnetic graphene oxide for water remediation. The subject of the paper is interesting. The materials characterization is broad. However, section of the Discussion contains some apodictic statements and some errors. This paper is suitable for the journal, but some parts should be improved.
MAJOR REVISION should follow the recommendations below.
A few specific comments:
p.4, Table 1: In the table at least two equations (9) and (10) must be corrected by the authors. Please check all the mathematical expressions.
A: Thanks for the observation. Corrected.
p.4, Table 1: the correct value of the universal gas constant R is 8.314 J mol-1 K-1
A: Corrected.
p.6, fig.2(b): Why is the GO peak (001) completely missing?
A: The graphene oxide surface was coated by magnetite nanoparticles (1). Nevertheless it is possible to observe in SEM results. The XRD discussion was improved (See Lines 167-169).
(1): Zeng, X. Bai, Y. Yang, H. Zhu, L. Yua, R. Solvothermal synthesis and good microwave absorbing properties for magnetic porous-Fe3O4/graphene nanocomposites. AIP Advances. 2017,7, 056605.
p.8, line 208: “Fig.5” should be “Fig.6”.
A: Thank you. Corrected.
p.9, line 214: Please, give an example where the GO-Fe3O4 composite has better physicochemical stability.
A: A previous study developed by our research group reported the high physicochemical stability of magnetic graphene oxide through an experiment using hydrochloric acid, sulfuric acid, and sodium hydroxide (2). The results showed that only using concentrated hydrochloric acid did the material lose its magnetic behavior.
(2) Rhoden, C.R.B.; Bruckmann, F. da S.; Salles, T. da R.; Kaufmann Junior, C.G.; Mortari, S.R. Study from the Influence of Magnetite onto Removal of Hydrochlorothiazide from Aqueous Solutions Applying Magnetic Graphene Oxide. J. Water Proc. Engineering 2021, 43, 102262. https://doi.org/10.1016/j.jwpe.2021.102262
p.9, line 228: “The removal value reduces by around 23% from the lowest to the highest concentration” the sentence should be “The removal value increases by around 23% from the lowest to the highest concentration”
A: Thank you for your observation. This misspelling was corrected.
p.11, fig.10: In the structural formula of CPT a CH3 is missing. Consider that the formula is C9H15NO3S.
A: Thank you for your observation. The structural formula of captopril was corrected.
p.12, line 309: It is important that the Elovich equation is an empirical function. Hence, the assignment of physical meaning to the value of its parameters alpha and beta is questionable.
A: In fact, the Elovich kinetic equation is a mathematical expression without physical meaning. However, the kinetic models are widely used to describe the kinetic behavior (kinetic curves), i.e., employed to verify the adsorption rate over time.
p.15 Table 4: Thermodynamic parameters are not consistent. Using the equation DG°=DH°-TDS° we get -6,2 kJ mol-1.
A: Thank you for the correction. The value was recalculated and changed in the main manuscript (See Table 4, Line 384).
- 15, Conclusions: In the conclusions, it is important to point out that the pH=3 conditions used in this experimental study are not reasonably applicable in a sewage treatment plant.
A: In fact, the experimental condition is not commonly applicable in wastewater treatment systems. However, this study was conducted in a wide pH range to verify the adsorption behavior under different conditions. Although pH 3.0 is the better condition, the other pH also showed an excellent adsorption efficiency, proving the potential applicability in waste management.
MINOR REVISION:
p.1, line 19: “captopril” should be “captopril (CPT)”
A: Corrected.
p.2, line79: “300 mL of distilled water…” the phrase is suspended (incomplete).
A: Corrected.
p.4, line at the bottom: “Sun squares error” should be “Sum squares error”
A: Corrected.
p.15, line371: “Accordin” should be “According”
A: Corrected.
Fig. 2(b): Why is the graphene oxide peak (001) totally absent?
A: The graphene oxide surface was coated by magnetite nanoparticles (1). Nevertheless it is possible to observe in SEM results. The XRD discussion was improved (Lines 167-169).
(1): Zeng, X. Bai, Y. Yang, H. Zhu, L. Yua, R. Solvothermal synthesis and good microwave absorbing properties for magnetic porous-Fe3O4/graphene nanocomposites. AIP Advances. , 2017,7, 056605.
p.4 line 208: “According to Fig. 5,” should be “According to Fig. 6”.
A: Thank you. Corrected.

Reviewer 2 Report
1)Abstract: CPT acronym should be place after the first mention of captopril .
2) It is recognized that R2 and adjusted R2 are not suitable for comparing non linear models (see https://doi.org/10.1186/1471-2210-10-6). Please use a more appropriate statistical method to compare Sips, Freundlich and Langmuir isotherms.
3) Sentences like “Thermodynamic parameters reveal a spontaneous, exothermic, involving a chemisorption process.” or “it was possible to assume that the captopril adsorption was a spontaneous and thermodynamically favorable process” are wrong. The sign of DG° is NOT related to the spontaneity of the process. This is a very common mistake in adsorption literature. Please correct the sentence as “Thermodynamic parameters reveal a FAVOURABLE, exothermic, involving a chemisorption process.” and “it was possible to infer that the captopril adsorption was a thermodynamically favorable process“; The following article clearly explain the above mentioned issue, add it to your manuscript’s references: https://doi.org/10.1016/j.molliq.2022.118762.
4) Which are the equilibrium constant values used in the van’t Hoff equation?
5) Where is the isotherms' plot?
6) All (or most of) the kinetic data seem to be at equilibrium. The adsorption kinetic data from the initial phase are completely missing. This could make difficult to obtain a reliable estimation of the fitted parameters, especially of the kinetic rate constants. Please provide the standard error associated to all the estimated parameters and do the same for the isotherm parameters.
7) Table 1: please change DG into DG°. Moreover, change “sun square errors” into “sum square errors”.
Author Response
We thank you for the opportunity to submit our revised paper to the Water. We revised the manuscript at all with great attention. The Referee's comments provide us with detailed and very useful reports. The modifications are highlighted in yellow color in the final version of the manuscript. By addressing their comments in the revised version, we are confident that the paper has been considerably improved, thus shaping it to a publishable form. Please, find below the answers to the referee's comments.
Reviewer 2
Authors: We thank the referee for our paper evaluation, and we consider the comments and suggestions to improve the paper a lot.
1)Abstract: CPT acronym should be place after the first mention of captopril.
A: Thanks for the observation. Corrected.
2) It is recognized that R2 and adjusted R2 are not suitable for comparing non linear models (see https://doi.org/10.1186/1471-2210-10-6). Please use a more appropriate statistical method to compare Sips, Freundlich and Langmuir isotherms.
A: Thank you for your observation. In the main manuscript was added MSE error for better comparison of nonlinear models.
3) Sentences like “Thermodynamic parameters reveal a spontaneous, exothermic, involving a chemisorption process.” or “it was possible to assume that the captopril adsorption was a spontaneous and thermodynamically favorable process” are wrong. The sign of DG° is NOT related to the spontaneity of the process. This is a very common mistake in adsorption literature. Please correct the sentence as “Thermodynamic parameters reveal a FAVOURABLE, exothermic, involving a chemisorption process.” and “it was possible to infer that the captopril adsorption was a thermodynamically favorable process“; The following article clearly explain the above mentioned issue, add it to your manuscript’s references: https://doi.org/10.1016/j.molliq.2022.118762.
A: Thanks for your observation. In fact, the sentences were corrected, and the reference was added.
4) Which are the equilibrium constant values used in the van’t Hoff equation?
A: Following the editor suggestions, the values of equilibrium were changed, removing the points that reached nearly to zero. (See Line 384).
5) Where is the isotherms' plot?
A: The isotherms plots were added (See lines 368-373).
6) All (or most of) the kinetic data seem to be at equilibrium. The adsorption kinetic data from the initial phase are completely missing. This could make difficult to obtain a reliable estimation of the fitted parameters, especially of the kinetic rate constants. Please provide the standard error associated to all the estimated parameters and do the same for the isotherm parameters.
A: Thank you for the observation. It will improve a lot the manuscript. The standard error from all parameters were added for Kinetic and Isotherms models.
7) Table 1: please change DG into DG°. Moreover, change “sun square errors” into “sum square errors”.
A: Thank you. Corrected.

Reviewer 3 Report
I have a few questions regarding the manuscript:
1) Can authors comment on why there is no signal corresponding to GO in XRD of GO‧Fe3O4?
2) Line 228, "The removal value reduces by around 23% from the lowest to the highest concentration". I'm not sure this statement is consistent with Figure 7. Clearly removal value is the lowest with lowest concentration.
3) Line 256, "it has an anionic character in this pH condition", From the context, the pH is 3.0 and if pKa of Captopril is around 3.7, pKa is higher than pH, in this case, conjugate base (anionic) is smaller in quantity than acid. And when pH is between 4-7, captopril should not have a cationic characteristic according to its pKa. Can the authors explain this part?
Author Response
We thank you for the opportunity to submit our revised paper to the Water. We revised the manuscript at all with great attention. The Referee's comments provide us with detailed and very useful reports. The modifications are highlighted in yellow color in the final version of the manuscript. By addressing their comments in the revised version, we are confident that the paper has been considerably improved, thus shaping it to a publishable form. Please, find below the answers to the referee's comments.
With kind regards
Prof. Dr. Cristiano Rodrigo Bohn Rhoden
Corresponding Author
Reviewer 3
I have a few questions regarding the manuscript:
Authors: We thank the referee for our paper evaluation, and we consider the comments and suggestions to improve the paper a lot.
Queries are given in black and answers in blue.
1) Can authors comment on why there is no signal corresponding to GO in XRD of GO‧Fe3O4?
A: The graphene oxide surface was coated by magnetite nanoparticles (1). Nevertheless it is possible to observe in SEM results. The XRD discussion was improved (Lines 167-169).
(1): Zeng, X. Bai, Y. Yang, H. Zhu, L. Yua, R. Solvothermal synthesis and good microwave absorbing properties for magnetic porous-Fe3O4/graphene nanocomposites. AIP Advances. , 2017,7, 056605.
2) Line 228, "The removal value reduces by around 23% from the lowest to the highest concentration". I'm not sure this statement is consistent with Figure 7. Clearly removal value is the lowest with lowest concentration.
A: Thank you for the correction. In fact, this statement is wrong. This sentence was removed from the text.
3) Line 256, "it has an anionic character in this pH condition", From the context, the pH is 3.0 and if pKa of Captopril is around 3.7, pKa is higher than pH, in this case, conjugate base (anionic) is smaller in quantity than acid. And when pH is between 4-7, captopril should not have a cationic characteristic according to its pKa. Can the authors explain this part?
A: Due to the cationic surface of magnetic graphene oxide in an acid medium and the pKa of captopril, the attractive forces are favored under this condition (pH 3.0). The captopril exhibit only an ionizable group (carboxylic acid) (Fig. 1). Therefore, this drug does not present a cationic behavior at pH 4-7.
(1): Pereira, A. V.; Garabeli, A. A.; Schunemann, G. D.; Borck, P. C. Determination of dissociation constant (Ka) of captopril and nimesulide: analytical chemistry experiments for undergraduate pharmacy. Quim Nova 2011, 34, 1656-1660.
Fig. 1. Chemical species of CPT

Reviewer 4 Report
In this article, the adsorption of captopril from an aqueous solution using graphene oxide-based magnetic nano adsorbent has been investigated and modeled. The current study has good quality, but it is necessary to make the following corrections before the final acceptance
1) Barring a few sentences in the text, the English language is fair. However, the text is not free from grammar errors. Ensure that your English manuscript is guaranteed free of language issues. In addition, the manuscript should be thoroughly checked for English corrections as there are some colloquial terms being used.
2) Please revise the introduction section. The introduction is lack sufficient background information, which is unable to give the reader detailed background knowledge and possible wide application of this study. The introduction needs to be more emphasized the research work with a detailed explanation of the whole process considering past, present, and future scope. It needs to be more emphasized in the research work with a detailed explanation of the whole process. Research gaps should be highlighted more clearly and future applications of this study should be added
3) In the introduction section, it is necessary to write materials about, the effects of organic pollutants on the environment, and the removal methods of pollutants. It is suggested that the introduction section be improved using the suggested articles:
-https://doi.org/10.1080/00986445.2021.1963960
-https://doi.org/10.1016/j.jece.2020.104506
-https://doi.org/10.1007/s41204-021-00173-6
-https://doi.org/10.1002/cphc.200500294
-https://doi.org/10.1108/PRT-02-2021-0019
-https://doi.org/10.1016/j.jclepro.2018.06.146
-https://doi.org/10.1007/s11356-022-21554-7
-https://doi.org/10.1002/elan.202060271
-https://doi.org/10.1016/j.molliq.2022.120866
4) In section 2 (Materials and Methods), it is suggested that the chemical substances with their purity percentage should also be presented.
5) In Figs. 7, 8, 9, 11, 12, and 13, the value of the Error bar should be provided.
6) Compare the QM results with other studies reported in the literature
7) The conclusion is quite weak comprising only general statements. Add key values of the results. Add some strong lines proving the importance and novelty of this study and the possible future applicability of this study.
Author Response
We thank you for the opportunity to submit our revised paper to the Water. We revised the manuscript at all with great attention. The Referee's comments provide us with detailed and very useful reports. The modifications are highlighted in yellow color in the final version of the manuscript. By addressing their comments in the revised version, we are confident that the paper has been considerably improved, thus shaping it to a publishable form. Please, find below the answers to the referee's comments.
With kind regards
Prof. Dr. Cristiano Rodrigo Bohn Rhoden
Corresponding Author
Reviewer 4
In this article, the adsorption of captopril from an aqueous solution using graphene oxide-based magnetic nano adsorbent has been investigated and modeled. The current study has good quality, but it is necessary to make the following corrections before the final acceptance
Authors: We thank the referee for the positive evaluation of our paper, and we consider all the comments and suggestions to improve the paper.
Queries are given in black and answers in blue.
1) Barring a few sentences in the text, the English language is fair. However, the text is not free from grammar errors. Ensure that your English manuscript is guaranteed free of language issues. In addition, the manuscript should be thoroughly checked for English corrections as there are some colloquial terms being used.
A: Thank you for the observation. The manuscript was entire revised and submitted to English revisor.
2) Please revise the introduction section. The introduction is lack sufficient background information, which is unable to give the reader detailed background knowledge and possible wide application of this study. The introduction needs to be more emphasized the research work with a detailed explanation of the whole process considering past, present, and future scope. It needs to be more emphasized in the research work with a detailed explanation of the whole process. Research gaps should be highlighted more clearly and future applications of this study should be added
A: This section was fully revised and rewritten (See main manuscript, Lines 34-79).
3) In the introduction section, it is necessary to write materials about, the effects of organic pollutants on the environment, and the removal methods of pollutants. It is suggested that the introduction section be improved using the suggested articles:
A: The Introduction was improved by adding the references as requested.
-https://doi.org/10.1080/00986445.2021.1963960
-https://doi.org/10.1016/j.jece.2020.104506
-https://doi.org/10.1007/s41204-021-00173-6
-https://doi.org/10.1002/cphc.200500294
-https://doi.org/10.1108/PRT-02-2021-0019
-https://doi.org/10.1016/j.jclepro.2018.06.146
-https://doi.org/10.1007/s11356-022-21554-7
-https://doi.org/10.1002/elan.202060271
-https://doi.org/10.1016/j.molliq.2022.120866
4) In section 2 (Materials and Methods), it is suggested that the chemical substances with their purity percentage should also be presented.
A: Thanks, or the observation. The purity percentages of all chemical reagents were added in this section.
5) In Figs. 7, 8, 9, 11, 12, and 13, the value of the Error bar should be provided.
A: Thank you. In all the figures the Error bars were added.
6) Compare the QM results with other studies reported in the literature
A: The comparison with qmax of other studies was reported in the main manuscript (See lines 385-398).
7) The conclusion is quite weak comprising only general statements. Add key values of the results. Add some strong lines proving the importance and novelty of this study and the possible future applicability of this study.
A: The conclusion was rewritten, improving the important aspects finding in the current work. (See main manuscript, Lines 411-429).

Round 2
Reviewer 1 Report
The Authors have improved and corrected the manuscript.
MINOR REVISION:
p.4, Table 1: In equation (9) the change in Gibbs free energy should be under standard DG° conditions. If in column 1 are reported DG°,DH° and DS°, also equation (10) should be under standard conditions.
Author Response
We thank you for the opportunity to submit our revised paper to the Water. We fulfilled the second requests and highlighted in yellow color in the final version of the manuscript. By addressing their comments in the secondo revised version, we are confident that the paper has been considerably improved, thus shaping it to a publishable form. Please, find below the answers to the referee's comments.
With kind regards
Prof. Dr. Cristiano Rodrigo Bohn Rhoden
Reviewer 1
The Authors have improved and corrected the manuscript.
MINOR REVISION:
p.4, Table 1: In equation (9) the change in Gibbs free energy should be under standard DG° conditions. If in column 1 are reported DG°, DH° and DS°, also equation (10) should be under standard conditions.
A: Thanks for the correction. Corrected.
Reviewer 2
Minor points remain to be fixed:
-In Table 1 replace G, DH and DS with DG°, DH° and DS°, respectively. The description of the Sips parameter qs should be the same of the Langmuir parameter qmax. In the Kd description write: thermodynamic equilibrium constant. In the MSE statistics description, replace the generic yexp and ypred with qe,exp and qe,pred, respectively.
A: Thanks for the correction. Corrected.
As concerns point #4 of the first round of revision, the authors’ response is not satisfying. The authors should clearly state in the manuscript how they got the equilibrium constants used for the van’t Hoff equation, and their values should be reported.
A: Thanks for the observation. The methodology and the values of the equilibrium constant were added in the main manuscript. (See Line 137-140 and Line 386, Table 4).

Reviewer 2 Report
Minor points remain to be fixed:
-In Table 1 replace DG, DH and DS with DG°, DH° and DS°, respectively. The description of the Sips parameter qs should be the same of the Langmuir parameter qmax. In the Kd description write: thermodynamic equilibrium constant. In the MSE statistics description, replace the generic yexp and ypred with qe,exp and qe,pred, respectively.
As concerns point #4 of the first round of revision, the authors’ response is not satisfying. The authors should clearly state in the manuscript how they got the equilibrium constants used for the van’t Hoff equation, and their values should be reported.
Author Response

(The authors gave the same response as above.)

Reviewer 4 Report
Accept
Author Response

(The authors gave the same response as above.)
